# The effect of non-linear signal in classification problems using gene expression

Benjamin J. Heil[1], Jake Crawford[1], Casey S. Greene[2,3]*

1 Genomics and Computational Biology Graduate Group, Perelman School of Medicine, University of Pennsylvania, Pennsylvania, United States of America, 2 Department of Pharmacology, University of Colorado School of Medicine, Colorado, United States of America, 3 Department of Biochemistry and Molecular Genetics, University of Colorado School of Medicine, Colorado, United States of America

* casey.s.greene@cuanschutz.edu

## Abstract

Those building predictive models from transcriptomic data are faced with two conflicting perspectives. The first, based on the inherent high dimensionality of biological systems, supposes that complex non-linear models such as neural networks will better match complex biological systems. The second, imagining that complex systems will still be well predicted by simple dividing lines prefers linear models that are easier to interpret. We compare multi-layer neural networks and logistic regression across multiple prediction tasks on GTEx and Recount3 datasets and find evidence in favor of both possibilities. We verified the presence of non-linear signal when predicting tissue and metadata sex labels from expression data by removing the predictive linear signal with Limma, and showed the removal ablated the performance of linear methods but not non-linear ones. However, we also found that the presence of non-linear signal was not necessarily sufficient for neural networks to outperform logistic regression. Our results demonstrate that while multi-layer neural networks may be useful for making predictions from gene expression data, including a linear baseline model is critical because while biological systems are high-dimensional, effective dividing lines for predictive models may not be.

## Author summary

If we could consistently predict biological conditions from mRNA levels, it could help discover biomarkers for disease diagnosis. Deep learning has become widely used for many tasks including biomarker discovery. It is unclear whether the complexity of these models is helpful. We evaluate whether or not more complex non-linear models have an advantage over simpler linear ones for a set of prediction tasks. We find that, at least for tissue prediction and prediction of metadata-derived sex prediction, linear models perform just as well as non-linear ones. However, we also demonstrate the presence of a predictive signal in the data that only the non-linear models can use. Our results suggest that the nonlinear signals may be redundant with linear ones or that current deep neural networks are not able to successfully use the signal when linear signals are present.

Data Availability Statement: The code, data, and model weights to reproduce this work can be found at https://github.com/greenelab/linear_signal.

**Funding:** This work was supported by grants from the National Institutes of Health's National Human Genome Research Institute (NHGRI) under award R01 HG010067 and the Gordon and Betty Moore Foundation (GBMF 4552) to CSG. The funders had no role in study design, data collection and analysis, decision to publish, or preparation of the manuscript.

**Competing interests:** The authors declare that they have no conflict of interest.

This is a *PLOS Computational Biology* Methods paper.

## Introduction

Transcriptomic data contains a wealth of information about biology. Gene expression-based models are already being used for subtyping cancer [1], predicting transplant rejections [2], and uncovering biases in public data [3]. In fact, both the capability of machine learning models [4] and the amount of transcriptomic data available [5,6] are increasing rapidly. It makes sense, then, that neural networks are frequently being used to build predictive models from transcriptomic data [7–9].

However, there are two conflicting ideas in the literature regarding the utility of non-linear models. One theory draws on prior biological understanding: the paths linking gene expression to phenotypes are complex [10,11], and non-linear models like neural networks should be more capable of learning that complexity. Unlike purely linear models such as logistic regression, non-linear models can learn non-linear decision boundaries to differentiate phenotypes. Accordingly, many have used non-linear models to learn representations useful for making predictions of phenotypes from gene expression [12–14].

The other supposes that even high-dimensional complex systems may have linear decision boundaries. This is supported empirically: linear models seem to do as well as or better than non-linear ones in many cases [15]. While papers of this sort are harder to come by—perhaps scientists do not tend to write papers about how their deep learning model was worse than logistic regression—other complex biological problems have also seen linear models prove equivalent to non-linear ones [16,17].

We design experiments to ablate linear signal and find merit to both hypotheses. We construct a system of binary and multiclass classification problems on the GTEx and Recount3 compendia [18,19] that shows linear and non-linear models have similar accuracy on several prediction tasks. However, when we remove any linear separability from the data, we find non-linear models are still able to make useful predictions even when the linear models previously outperformed the non-linear ones. Given the unexpected nature of these findings, we evaluate independent tasks, examine different problem formulations, and verify our models' behavior with simulated data. The models' results are consistent across each setting, and the models themselves are comparable, as they use the same training and hyperparameter optimization processes [20].

In reconciling these two ostensibly conflicting theories, we confirm the importance of implementing and optimizing a linear baseline model before deploying a complex non-linear approach. While non-linear models may outperform simpler models at the limit of infinite data, they do not necessarily do so even when trained on the largest datasets publicly available today.

## Results

### Linear and non-linear models have similar performance in many tasks

We compared the performance of linear and non-linear models across multiple datasets and tasks (Fig 1A). We examined using TPM-normalized RNA-seq data to predict tissue labels from GTEx [18], tissue labels from Recount3 [19], and metadata-derived sex labels from Flynn et al. [3]. To avoid leakage between cross-validation folds, we placed entire studies into single folds (Fig 1B). We evaluated models on subsampled datasets to determine the extent to which performance was affected by the amount of training data.

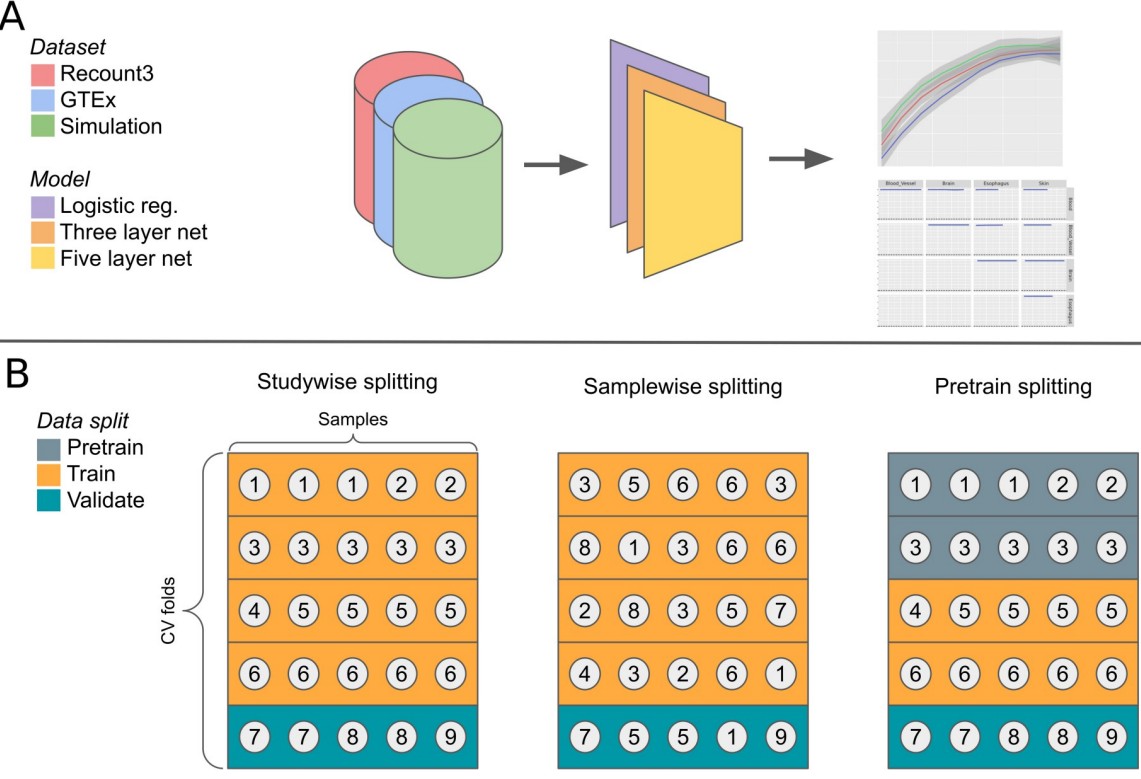

**Fig 1. Schematic of the model analysis workflow.** We evaluate three models on multiple classification problems in three datasets (A). We stratify the samples into cross-validation folds based on their study (in Recount3) or donor (in GTEx). We also evaluate the effects of sample-wise splitting and pretraining (B). Each circle in the figure represents a sample, which is numbered according to its study.

We used GTEx [18] to determine whether linear and non-linear models performed similarly on a well-characterized dataset with consistent experimental protocols across samples. We first trained our models to differentiate between tissue types on pairs of the five most common tissues in the dataset. Likely due to the clean nature of the data, all models were able to perform perfectly on these binary classification tasks (Fig 2A). Because binary classification was unable to differentiate between models, we evaluated the models on a more challenging task. We tested the models on their ability to perform multiclass classification on all 31 tissues

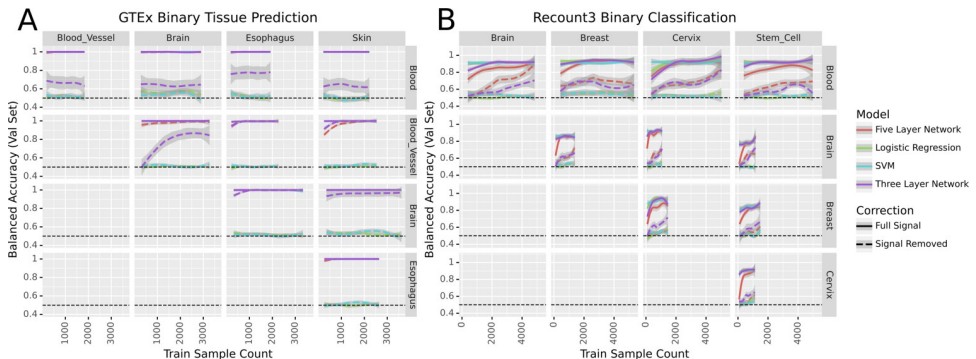

**Fig 2. Models' performance across binary classification tasks before and after signal removal in the Recount and GTEx datasets.**

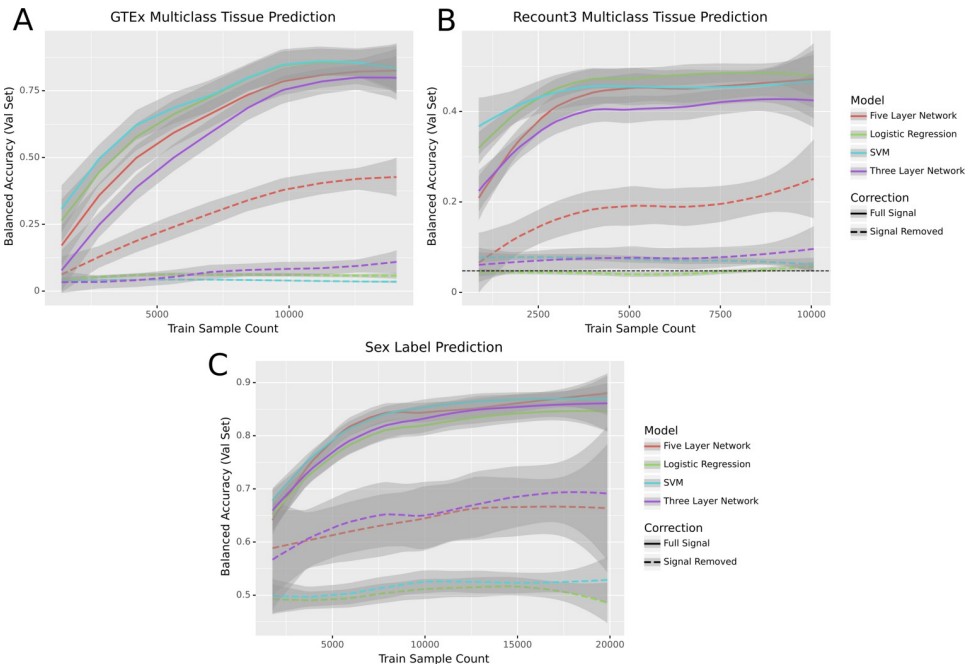

**Fig 3. Performance of models across three classification tasks before and after signal removal.** In each panel the loess curve and its 95% confidence interval are plotted based on points from three seeds, ten data subsets, and five folds of study-wise cross-validation (for a total of 150 points per model per panel). It is worth noting that "Sample Count" in these figures refers to the total number of RNA-seq samples, some of which share donors. As a result, the effective sample size may be lower than the sample count.

present in the dataset. In the multitask setting, both the linear support vector machine (SVM) and logistic regression slightly outperformed the five-layer neural network, which in turn slightly outperformed the three-layer net (Fig 3A).

We then evaluated the same approaches in a dataset with very different characteristics: Sequence Read Archive [21] samples from Recount3 [19]. We compared the models' ability to differentiate between pairs of tissues (Fig 2B) and found their performance was roughly equivalent. We also evaluated the models' performance on a multiclass classification problem differentiating between the 21 most common tissues in the dataset. As in the GTEx setting, the linear models outperformed the five-layer network, which outperformed the three-layer network (Fig 3B).

To examine whether these results held in a problem domain other than tissue type prediction, we tested performance on metadata-derived sex labels (Fig 3C), a task previously studied by Flynn et al. [3]. We used the same experimental setup as in our other binary prediction tasks to train the models, but rather than using tissue labels we used sex labels from Flynn et al. In this setting we found that while the models all performed similarly, the non-linear models tended to have a slight edge over the linear ones.

It is worth noting that the relative performance of the models remain the same despite class imbalances and different ratios of genes to samples. For example, the ratio of genes to samples goes all the way from roughly 10:1 in the heavily subsetted binary classification regime to 1:4 in the full sex label prediction dataset.

## There is predictive non-linear signal in transcriptomic data

Our results to this point are consistent with a world where the predictive signal present in transcriptomic data is entirely linear. If that were the case, non-linear models like neural networks

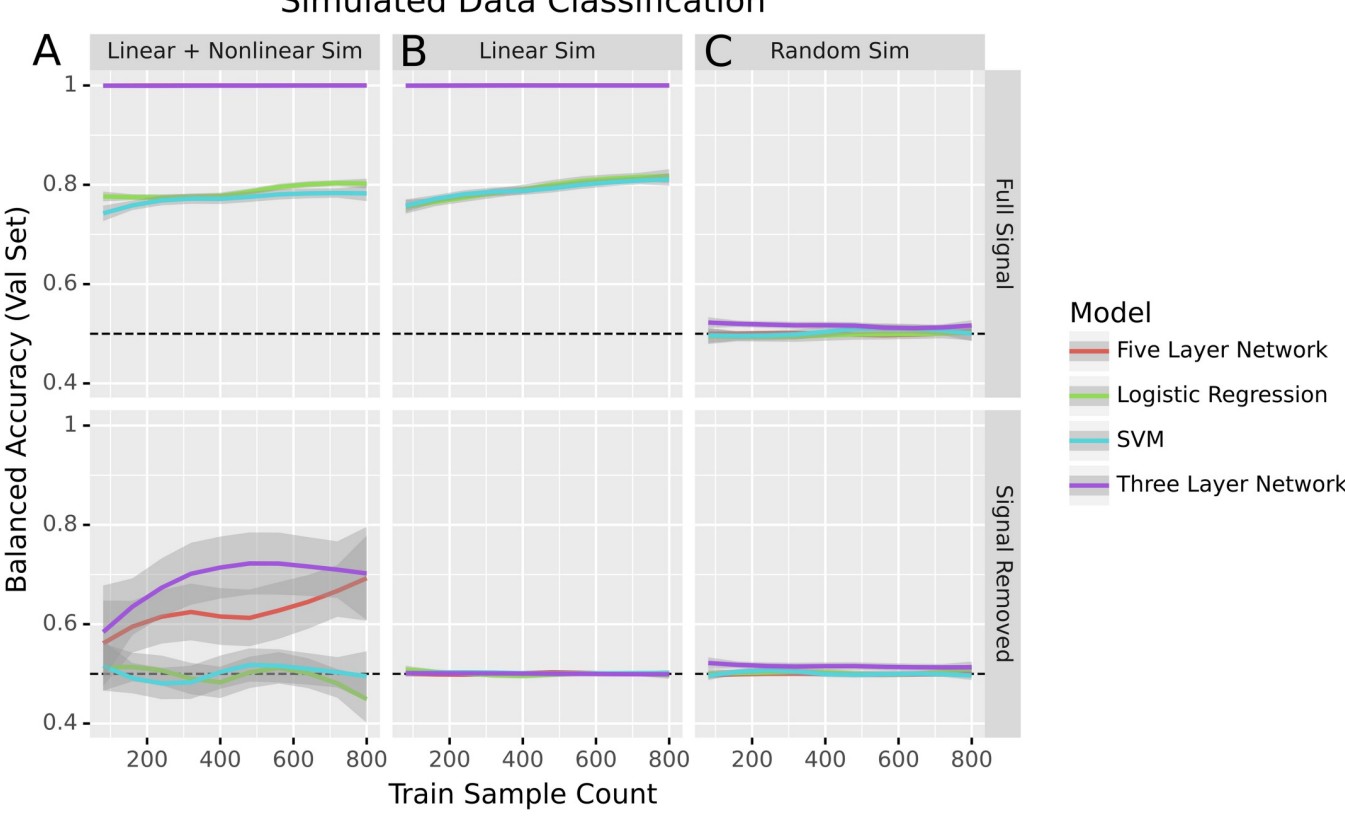

**Fig 4. Performance of models in binary classification of simulated data before and after signal removal.** Dotted lines indicate expected performance for a naive baseline classifier that predicts the most frequent class.

would fail to give any substantial advantage. However, based on past results we expect there to be relevant non-linear biological signal [22]. To get a clearer idea of what that would look like, we simulated three datasets to better understand model performance for a variety of data generating processes. We created data with both linear and non-linear signal by generating two types of features: half of the features with a linear decision boundary between the simulated classes and half with a non-linear decision boundary (see Methods for more details). After training to classify the simulated dataset, all models effectively predicted the simulated classes. To determine whether or not there was non-linear signal, we then used Limma [23] to remove the linear signal associated with the endpoint being predicted. After removing the linear signal from the dataset, non-linear models correctly predicted classes, but logistic regression and the SVM performed no better than random (Fig 4A).

To confirm that non-linear signal was key to the performance of non-linear methods, we generated another simulated dataset consisting solely of features with a linear decision boundary between the classes. As before, all models were able to predict the different classes well. However, once the linear signal was removed, all models performed no better than random guessing (Fig 4B). That the non-linear models only achieved baseline accuracy also indicated that the signal removal method was not injecting non-linear signal into data where non-linear signal did not exist.

We also trained the models on a dataset where all features were Gaussian noise as a negative control. As expected, the models all performed at baseline accuracy both before and after the signal removal process (Fig 4C). This experiment supported our decision to perform signal

removal on the training and validation sets separately. One potential failure state when using the signal removal method would be if it induced new signal as it removed the old. Such a state can be seen when removing the linear signal in the full dataset (S1 Fig).

We next removed linear signal from GTEx and Recount3. We found that the neural nets performed better than the baseline while logistic regression and the SVM did not (Figs 3 and 2). For multiclass problems the linear models performed poorly while the non-linear models had performance that increased with an increase in data while remaining worse than before the linear signal was removed (Fig 3A and 3B) Likewise, the sex label prediction task showed a marked difference between the neural networks and the linear models: only the neural networks could learn from the data (Fig 3C). In each of the settings, the models performed less well when run on data with signal removed, indicating an increase in the problem's difficulty. Logistic regression and the linear SVM, in particular, performed no better than random.

To verify that our results were not an artifact of our decision to assign studies to cross-validation folds rather than samples, we compared the study-wise splitting that we used with an alternate method called sample-wise splitting. Sample-wise splitting (see Methods) is common in machine learning, but can leak information between the training and validation sets when samples are not independently and identically distributed among studies—a common feature of data in biology [24]. We found that sample-wise splitting induced substantial performance inflation (S2 Fig). The relative performance of each model stayed the same regardless of the data splitting technique, so the results observed were not dependent on the choice of splitting technique.

We also ensured that these results were also applicable at small sample sizes by running the GTEx binary prediction tasks with the training dataset reduced to 1–10% of its original size. We continued to see similar performance across model classes, though the accuracies were lower across the board due to the decreased training data (S3 Fig).

Another growing strategy in machine learning, especially on biological data where samples are limited, is training models on a general-purpose dataset and fine-tuning them on a dataset of interest. We examined the performance of models with and without pretraining (S4 Fig). We split the Recount3 data into three sets: pretraining, training, and validation (Fig 1B), then trained two identically initialized copies of each model. One was trained solely on the training data, while the other was trained on the pretraining data and fine-tuned on the training data. The pretrained models showed a gain in performance over the non-pretrained models even when trained with small amounts of data from the training set. However, the non-linear models did not have a greater increase in performance from pretraining than the linear models, and the balanced accuracy was similar across pretrained models.

## Methods

### Datasets

**GTEx.** We downloaded the 17,382 TPM-normalized samples of bulk RNA-seq expression data available from version 8 of GTEx. We zero-one standardized the data and retained the 5000 most variable genes. The tissue labels we used for the GTEx dataset were derived from the 'SMTS' column of the sample metadata file.

**Recount3.** We downloaded RNA-seq data from the Recount3 compendium [25] during the week of March 14, 2022. Before filtering, the dataset contained 317,258 samples, each containing 63,856 genes. To filter out single-cell data, we removed all samples with greater than 75 percent sparsity. We also removed all samples marked 'scrna-seq' by Recount3's pattern matching method (stored in the metadata as 'recount_pred.pattern.predict.type'). We then converted the data to transcripts per kilobase million using gene lengths from BioMart [26]

and performed standardization to scale each gene's range from zero to one. We kept the 5,000 most variable genes within the dataset.

We labeled samples with their corresponding tissues using the 'recount_pred.curated.tissue' field in the Recount3 metadata. These labels were based on manual curation by the Recount3 authors. A total of 20,324 samples in the dataset had corresponding tissue labels. Samples were also labeled with their corresponding sex using labels from Flynn et al. [3]. These labels were derived using pattern matching on metadata from the European Nucleotide Archive [27]. A total of 23,525 samples in our dataset had sex labels.

**Data simulation.** We generated three simulated datasets. The first dataset contained 1,000 samples of 5,000 features corresponding to two classes. Of those features, 2,500 contained linear signal. That is to say that the feature values corresponding to one class were drawn from a standard normal distribution, while the feature values corresponding to the other were drawn from a Gaussian with a mean of 6 and unit variance.

We generated the non-linear features similarly. The values for the non-linear features were drawn from a standard normal distribution for one class, while the second class had values drawn from either a mean six or negative six Gaussian with equal probability. These features are referred to as "non-linear" because two dividing lines are necessary to perfectly classify such data, while a linear classifier can only draw one such line per feature.

The second dataset was similar to the first dataset, but it consisted solely of 2,500 linear features. The final dataset contained only values drawn from a standard normal distribution regardless of class label.

## Model architectures

We used three representative models to demonstrate the performance profiles of different model classes. The first was a linear model, ridge logistic regression, selected as a simple linear baseline to compare the non-linear models against. The next model was a three-layer fully-connected neural network with ReLU non-linearities [28] and hidden layers of size 2500 and 1250. This network served as a model of intermediate complexity: it was capable of learning non-linear decision boundaries, but not the more complex representations a deeper model might learn. Finally, we built a five-layer neural network to serve as a (somewhat) deep neural net. This model also used ReLU non-linearities, and had hidden layers of sizes 2500, 2500, 2500, and 1250. The five-layer network, while not particularly deep compared to, e.g., state of the art computer vision models, was still in the domain where more complex representations could be learned, and vanishing gradients had to be accounted for.

## Model training

We trained our models via a maximum of 50 epochs of mini-batch stochastic gradient descent in PyTorch [29]. Our models minimized the cross-entropy loss using an Adam [30] optimizer. They also used inverse frequency weighting to avoid giving more weight to more common classes. To regularize the models, we used early stopping and gradient clipping during the training process. The only training differences between the models were that the two neural nets used dropout [31] with a probability of 0.5, and the deeper network used batch normalization [32] to mitigate the vanishing gradient problem.

We ensured the results were deterministic by setting the Python, NumPy, and PyTorch random seeds for each run, as well as setting the PyTorch backends to deterministic and disabling the benchmark mode. The learning rate and weight decay hyperparameters for each model were selected via nested cross-validation over the training folds at runtime, and we tracked and recorded our model training progress using Neptune [33]. The one exception to this was

the Pytorch SVM model for sex prediction where weight regularization was removed to prevent training instability. Our work meets the bronze standard of reproducibility [34] and fulfills aspects of the silver and gold standards including deterministic operation and an automated analysis pipeline.

We also used Limma [23] to remove linear signal associated with tissues in the data. We ran the 'removeBatchEffect' function on the training and validation sets separately, using the tissue labels as batch labels. This function fits a linear model that learns to predict the training data from the batch labels, and uses that model to regress out the linear signal within the training data that is predictive of the batch labels.

## Model evaluation

In our analyses we used five-fold cross-validation with study-wise data splitting. In a study-wise split, the studies are randomly assigned to cross-validation folds such that all samples in a given study end up in a single fold (Fig 1B).

**Hardware.**   Our analyses were performed on an Ubuntu 18.04 machine and the Colorado Summit compute cluster. The desktop CPU used was an AMD Ryzen 7 3800xt processor with 16 cores and access to 64 GB of RAM, and the desktop GPU used was an Nvidia RTX 3090. The Summit cluster used Intel Xeon E5-2680 CPUs and NVidia Tesla K80 GPUs. From initiating data download to finishing all analyses and generating all figures, the full Snakemake [35] pipeline took around one month to run.

**Recount3 tissue prediction.**   In the Recount3 setting, the multi-tissue classification analyses were trained on the 21 tissues (see Supp. text 1) that had at least ten studies in the dataset. Each model was trained to determine which of the 21 tissues a given expression sample corresponded to.

To address class imbalance, our models' performance was then measured based on the balanced accuracy across all classes. Unlike raw accuracy, balanced accuracy (the mean across all classes of the per-class recall) isn't predominantly determined by performance on the largest class in an imbalanced class setting. For example, in a binary classification setting with 9 instances of class A and 1 instance of class B, successfully predicting 8 of the 9 instances of class A and none of class B yields an accuracy of 0.8 and a balanced accuracy of 0.44.

The binary classification setting was similar to the multiclass one. The five tissues with the most studies (brain, blood, breast, stem cell, and cervix) were compared against each other pairwise. The expression used in this setting was the set of samples labeled as one of the two tissues being compared.

The data for both settings were split in a stratified manner based on their study.

**GTEx classification.**   The multi-tissue classification analysis for GTEx used all 31 tissues. The multiclass and binary settings were formulated and evaluated in the same way as in the Recount3 data. However, rather than being split study-wise, the cross-validation splits were stratified according to the samples' donors.

**Simulated data classification/sex prediction.**   The sex prediction and simulated data classification tasks were solely binary. Both settings used balanced accuracy, as in the Recount3 and GTEx problems.

**Pretraining.**   When testing the effects of pretraining on the different model types, we split the data into three sets. Approximately forty percent of the data went into the pretraining set, forty percent went into the training set, and twenty percent went into the validation set. The data was split such that each study's samples were in only one of the three sets to simulate the real-world scenario where a model is trained on publicly available data and then fine-tuned on a dataset of interest.

To ensure the results were comparable, we made two copies of each model with the same weight initialization. The first copy was trained solely on the training data, while the second was trained on the pretraining data, then the training data. Both models were then evaluated on the validation set. This process was repeated four more times with different studies assigned to the pretraining, training, and validation sets.

## Discussion and conclusion

We performed a series of analyses to determine the relative performance of linear and non-linear models across multiple tasks. Consistent with previous papers [15,16], linear and non-linear models performed roughly equivalently in a number of tasks. That is to say that there are some tasks where linear models perform better, some tasks where non-linear models have better performance, and some tasks where both model types are equivalent.

However, when we removed all linear signal in the data, we found that residual non-linear signal remained. This was true in simulated data as well as GTEx and Recount3 data across several tasks. These results also held in altered problem settings, such as using a pretraining dataset before the training dataset and using sample-wise data splitting instead of study-wise splitting. This consistent presence of non-linear signal demonstrated that the similarity in performance across model types was not due to our problem domains having solely linear signals.

One limitation of our study is that the results likely do not hold in an infinite data setting. Deep learning models have been shown to solve complex problems in biology and tend to significantly outperform linear models when given enough data. However, we do not yet live in a world in which millions of well-annotated examples are available in many areas of biology. Our results are generated on some of the largest labeled expression datasets in existence (Recount3 and GTEx), but our tens of thousands of samples are far from the millions or billions used in deep learning research.

We are also unable to make claims about all problem domains or model classes. There are many potential transcriptomic prediction tasks and many datasets to perform them on. While we show that non-linear signal is not always helpful in tissue or sex prediction, and others have shown the same for various disease prediction tasks, there may be problems where non-linear signal is more important. It is also possible that other classes of models, be they simpler non-linear models or different neural network topologies, are more capable of taking advantage of the non-linear signal present in the data.

Ultimately, our results show that task-relevant non-linear signal in the data, which we confirm is present, does not necessarily lead non-linear models to outperform linear ones. Additionally, our results suggest that scientists making predictions from expression data should always include simple linear models as a baseline to determine whether more complex models are warranted.

## Supporting information

**S1 Text. Recount3 tissues used.**
(DOCX)

**S1 Fig. Full dataset signal removal in a dataset without signal.**
(TIFF)

**S2 Fig. Performance of Recount3 multiclass prediction with samplewise train/val splitting.**
(TIFF)

**S3 Fig. Performance of models on a binary prediction task with very limited training data.**
(TIFF)

**S4 Fig. Performance of Recount3 multiclass prediction with pretraining.**
(TIFF)

## Acknowledgments

We would like to thank Alexandra Lee for reviewing code that went into this project. We would also like to thank the past and present members of GreeneLab who gave feedback on this project during lab meetings. This work utilized resources from the University of Colorado Boulder Research Computing Group, which is supported by the National Science Foundation (awards ACI-1532235 and ACI-1532236), the University of Colorado Boulder, and Colorado State University.

## Author Contributions

**Conceptualization:** Benjamin J. Heil, Casey S. Greene.

**Formal analysis:** Benjamin J. Heil.

**Funding acquisition:** Casey S. Greene.

**Investigation:** Benjamin J. Heil.

**Methodology:** Benjamin J. Heil.

**Project administration:** Casey S. Greene.

**Resources:** Benjamin J. Heil, Jake Crawford.

**Software:** Benjamin J. Heil.

**Supervision:** Casey S. Greene.

**Visualization:** Benjamin J. Heil.

**Writing – original draft:** Benjamin J. Heil.

**Writing – review & editing:** Jake Crawford, Casey S. Greene.

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
