## [Decision Letter · Decision Letter 0]

5 Dec 2022

Dear Dr. Greene,

Thank you very much for submitting your manuscript "The Effect of Non-Linear Signal in Classification Problems using Gene Expression" for consideration at PLOS Computational Biology.

As with all papers reviewed by the journal, your manuscript was reviewed by members of the editorial board and by several independent reviewers. In light of the reviews (below this email), we would like to invite the resubmission of a significantly-revised version that takes into account the reviewers' comments.

We cannot make any decision about publication until we have seen the revised manuscript and your response to the reviewers' comments. Your revised manuscript is also likely to be sent to reviewers for further evaluation.

Sincerely,

Jie Liu

Academic Editor

PLOS Computational Biology

Lucy Houghton

Staff

PLOS Computational Biology

Reviewer's Responses to Questions

**Comments to the Authors:**

Reviewer #1: In this resubmitted manuscript, the authors systematically compare multi-layer neural networks and logistic regression across multiple prediction tasks on GTEx and Recount3 datasets. The authors verified the presence of non-linear signal when predicting tissue and metadata sex labels from expression data by removing the predictive linear signals. The authors also showed that the presence of non-linear signal was not necessarily sufficient for neural networks to outperform logistic regression. Finally, the authors argued that while multi-layer neural networks might be useful for making predictions from gene expression data, including a linear baseline model is critical.

The paper is very well-written and easy to follow. The experiments are well designed, and the different stratification strategy of the data splitting is thoughtful. I don’t have any comments because they have been already well addressed in the response!

Reviewer #2: The authors try to evaluate the performance of linear and non-linear models in the presence and absence of linear signals. Logistic Regression was taken as a representative of linear models whereas a three-layered and a five-layered neural network were considered as non-linear models. Based on the experimental results in binary and multi-class classification tasks on the GTEx, Recount3, and three simulated datasets, the authors claim that the presence of task-related non-linear signals in the data does not necessarily assist the non-linear models to outperform the linear ones. Although the authors focus on an interesting topic and perform several experiments, still a lot more experiments with varying factors needs to be done before reaching any reasonable conclusion.

Strengths:

• Through experimentation by removing the linear signals, the authors prove that the similarity in the performance of the different models was not due to the sole presence of linear signals.

• Assessment of the models’ performance was also done with pretraining where the balanced accuracy was similar across the model.

• Experiments with sample-wise splitting and study-wise splitting better justify the results and uphold the claim of the authors.

Drawbacks:

• It would be interesting to assess the models’ performance in the case of small imbalanced datasets with high dimensionality which is often the case in biological data.

• More sophisticated linear models like- SVM could be used for comparison. Similarly, deeper models with hundred layers would be better representative of non-linear models.

• Experiments by varying the number of genes for the specific tasks would be helpful in the evaluation of the authors’ claim.

Reviewer #3: Some suggestions and questions:

1. Although the reviewers in the previous round pointed it out, I am still confused by the number in Fig 1B.

2. Figure numbers are not cited in the order. For example, Fig4A appeared before Fig2 and 3.

3. Above Figure 2, there are three words "TODO update description." What are they?

4. There is no supp. fig.4B, but it was cited above the Figure 2.

5. Why does the logistic regression perform much worse than the other two non-linear models under the linear simulation scenario (Figure 3A)? Also, the logistic regression model performs similarly for the linear and non-linear simulation (Figures 3A and B). Is it a little bit counterintuitive?

6. In the last sentence before the Methods section: "However, the non-linear models did not have a greater performance gain from pretraining than logistic regression, and the balanced accuracy was similar across models." If it is referring to supp.fig.8, I did see a big improvement for the non-linear model, and balanced accuracy definitely varies across models. I am curious how did the authors get this conclusion?

**Have the authors made all data and (if applicable) computational code underlying the findings in their manuscript fully available?**

Reviewer #1: Yes

Reviewer #2: Yes

Reviewer #3: Yes

PLOS authors have the option to publish the peer review history of their article (what does this mean?). If published, this will include your full peer review and any attached files.

Reviewer #1: No

Reviewer #2: No

Reviewer #3: No
---

## [Decision Letter · Decision Letter 1]

28 Feb 2023

Dear Dr. Greene,

We are pleased to inform you that your manuscript 'The Effect of Non-Linear Signal in Classification Problems using Gene Expression' has been provisionally accepted for publication in PLOS Computational Biology.

Best regards,

Jie Liu

Academic Editor

PLOS Computational Biology

Lucy Houghton

Staff

PLOS Computational Biology

Reviewer's Responses to Questions

**Comments to the Authors:**

Reviewer #1: I have no further comments

Reviewer #2: The Authors have addressed all of my concerns with the original manuscript. The revised manuscript is ready for publication.

Reviewer #3: Authors have addressed all my questions.

**Have the authors made all data and (if applicable) computational code underlying the findings in their manuscript fully available?**

Reviewer #1: None

Reviewer #2: Yes

Reviewer #3: None

PLOS authors have the option to publish the peer review history of their article (what does this mean?). If published, this will include your full peer review and any attached files.

Reviewer #1: No

Reviewer #2: No

Reviewer #3: No

---

## [Editor Report · Acceptance letter]

17 Mar 2023

PCOMPBIOL-D-22-01480R1 

The Effect of Non-Linear Signal in Classification Problems using Gene Expression

Dear Dr Greene,

I am pleased to inform you that your manuscript has been formally accepted for publication in PLOS Computational Biology. Your manuscript is now with our production department and you will be notified of the publication date in due course.

With kind regards,

Zsofia Freund
